# The Effects Induced by Spinal Manipulative Therapy on the Immune and Endocrine Systems

**DOI:** 10.3390/medicina55080448

**Published:** 2019-08-07

**Authors:** Andrea Colombi, Marco Testa

**Affiliations:** Department of Neuroscience, Rehabilitation, Ophtalmology, Genetics, Maternal and Child Health, University of Genova, Campus of Savona, Via Magliotto, 2 17100 Savona, Italy

**Keywords:** spinal manipulative therapy, immune system, endocrine system, back pain, neck pain, physiotherapy

## Abstract

*Background and Objectives*: Spinal manipulations are interventions widely used by different healthcare professionals for the management of musculoskeletal (MSK) disorders. While previous theoretical principles focused predominantly on biomechanical accounts, recent models propose that the observed pain modulatory effects of this form of manual therapy may be the result of more complex mechanisms. It has been suggested that other phenomena like neurophysiological responses and the activation of the immune-endocrine system may explain variability in pain inhibition after the administration of spinal manipulative therapy (SMT). The aim of this paper is to provide an overview of the available evidence supporting the biological plausibility of high-velocity, low-amplitude thrust (HVLAT) on the immune-endocrine system. *Materials and Methods*: Narrative critical review. An electronic search on MEDLINE, ProQUEST, and Google Scholar followed by a hand and “snowballing” search were conducted to find relevant articles. Studies were included if they evaluated the effects of HVLAT on participants’ biomarkers. *Results*: The electronic search retrieved 13 relevant articles and two themes of discussion were developed. Nine studies investigated the effects of SMT on cortisol levels and five of them were conducted on symptomatic populations. Four studies examined the effects of SMT on the immune system and all of them were conducted on healthy individuals. *Conclusions*: Although spinal manipulations seem to trigger the activation of the neuroimmunoendocrine system, the evidence supporting a biological account for the application of HVLAT in clinical practice is mixed and conflicting. Further research on subjects with spinal MSK conditions with larger sample sizes are needed to obtain more insights about the biological effects of spinal manipulative therapy.

## 1. Introduction

Spinal disorders are conditions affecting an increasing number of people, and their associated direct and indirect costs have dramatically grown around the globe [1]. The 12-month incidence of neck pain ranges between 10.4% and 21.3% [2], and more than 70% of people living in Western countries experience an episode of lower back pain in their lifetime [3].

Although economic and societal systems are striving to minimize the expenditures associated with their management, the US health care system alone devotes more than 80 billions of dollars per annum for the treatment of these clinical entities [4]. Significant efforts are made by pain research to find effective interventions, yet these musculoskeletal conditions remain the leading causes for years lived with disabilities (YLDs) in most countries and territories. The Global Burden of Disease 2016 study reports that both the prevalence and the YLDs because of low back and neck pain have increased by more than 18% over the last decade [5]. 

Although the overall literature suggests that a significant proportion of spinal symptoms are unlikely to reflect any serious disease and are mostly self-limited, a small minority of patients progress to a chronic, life-changing state [6]. This transition to persistent pain can be associated with metabolic, anatomical, and functional disruption, and it can affect multiple dimensions such as cognitive, emotional, and behavioral responses [7]. 

Although a number of pharmacologic and non-pharmacologic interventions have been recommended by international guidelines, their implementation in clinical practice seems less than optimal [8].

Thus, there is growing consensus indicating the urgent need for clinicians to more strictly adhere to these recommendations in the management of such musculoskeletal (MSK) disorders [9].

For example, regarding lower back pain, a great emphasis is placed on education and self-care as first-line interventions. Second-line options include, but are not limited to, analgesics, physical therapy, exercise, and cognitive therapies [10]. 

Among secondary options, spinal manipulative therapy (SMT), also known as high-velocity, low-amplitude thrust (HVLAT), is a ‘hands-on’ approach classified as an adjunctive strategy commonly implemented by physiotherapists, osteopaths, and chiropractors [11].

Recommendations around SMT can vary substantially among countries and depending on the disorder to be treated. For example, in the UK [12], for the treatment of lower back pain and sciatica, SMT is suggested only if incorporated as part of a multimodal treatment including exercise.

The International Federation of Orthopaedic Manual Physical Therapists (IFOMPT) defines manipulations as “a passive, high velocity, low amplitude thrust applied to a joint complex within its anatomical limit (active and passive motion occurs within the range of motion of the joint complex and not beyond the joint’s anatomic limit) with the intent to restore optimal motion, function, and/ or to reduce pain” [13].

For many years, the leading theories underlying the effects of SMT on pain modulation were purely mechanistic-based. Much of the literature focused on specific mechanical mechanisms of SMT, such as its inherent capacity to realign joint “subluxations”, to restore dysfunctional vertebral segments or release of “locked” zygapophyseal joints [14]. 

During the past 20 years, however, accumulating evidence has challenged these earlier assertions and much more information has become available supporting the role of the neuro-immune-endocrine system and other non-specific factors (contextual/psychological) accounting for variations in clinical outcomes [15,16]. High-quality systematic reviews report that SMT leads to minimal and short-term clinical benefits in terms of pain relief and functionality for the treatment of lower back [17] and neck pain [18]. Yet, a detailed understanding of the underpinning biological mechanisms still remains elusive. 

A recent systematic review with meta-analysis [19] has quantitatively summarized the evidence regarding the SMT-induced effects on the neuro-immune-endocrine system. In particular, researchers aimed to quantify changes in levels of circulating neuropeptides, inflammatory and endocrine biomarkers with samples collected from any body fluids (blood/urine/saliva).

However, due to the nature of their study design and stringent inclusion criteria, little attention may have been given to crucial methodological issues that can affect the evaluation of the effects of spinal manipulations. For example, aspects such as recruitment methods, the design of interventions and control groups, the effect of contextual factors or biomarkers measurement may play a crucial role [20,21,22].

Therefore, the purpose of this paper is not to provide a systematic review or meta-analytic approach to the efficacy of HVLAT, but instead to overview how and why these interventions might exert a neuro-immune-endocrine effect on the human body.

## 2. Materials and Methods

Literature was identified through an electronic search on databases such as MEDLINE, ProQUEST, and Google Scholar to retrieve papers, dissertations, and theses investigating the effects of HVLAT on the immune and endocrine system. Articles were considered if they included healthy people or patients suffering from spinal pain to whom it was delivered HVLAT as intervention. Acceptable control groups included “sham” manipulation, joint mobilization, hands positioning, or no intervention. A broad search was conducted with a combination of MeSH terms and keywords such as “spinal manipulation”, “vertebral manipulation”, “spinal manipulative therapy”, “endocrine system”, “biomarkers”, and “immune system”. The search strategy can be found in Appendix A. Next, reference lists of retrieved articles, previous review articles on the topic, and manual searches were conducted in the databases and journals for authors who regularly publish in this area.

## 3. Results

The literature search on MEDLINE, Google Scholar, and ProQuest retrieved 1707 records and nine articles were included as they met the inclusion criteria [23,24,25,26,27,28,29,30,31]. The manual search yielded four additional manuscripts totaling 13 relevant articles. Two themes of discussion, namely the effect of HVLAT on endocrine and immune system markers, were built upon the selected articles. Nine studies investigated the effects of SMT on cortisol levels [24,25,26,30,31,32,33,34,35] and five of them were conducted on a symptomatic population [24,25,33,34,35]. Five studies examined cortisol levels through saliva samples [30,31,32,34,35], whereas four used the venepuncture technique [24,25,26,33]. Four studies examined the effects of SMT on the immune system and all of them were conducted on healthy individuals [23,27,28,29]. 

## 4. Discussion

### 4.1. Neuro-Immune-Endocrine Effects

In recent years, the central role that a dysfunctional neuro-immune-endocrine system may play in musculoskeletal (MSK) pain has gained considerable attention [36,37]. It is suggested that low-grade inflammation may be associated with the severity of lower back (LBP) and neck pain (NP) as systemically elevated pro-inflammatory cytokines and chemokines have been found in these patients [38,39].

These neurochemicals including but not limited to, tumor necrosis factor (TNF-α), interleukin-1 (IL-1), interleukin-6 (IL-6), or interleukin-8 (IL-8), have been consistently revealed in cohorts of patients presenting with joints degeneration of the vertebral column [40,41,42].

Indeed, structural modifications occurring in cartilaginous endplates and vertebral bodies may be the consequence of augmented infiltration of immune cells at these sites and not just the result of ageing [43]. However, whether these biochemical and cellular changes are the leading cause of structural abnormalities or if tissue failure is responsible for their production has not been fully elucidated.

Additionally, such low-grade inflammation may be the result of an impaired cortisol regulation, which has been found in patients with chronic low back pain [44], fibromyalgia [45], and temporomandibular disorders [46], among others [47,48].

Cortisol is a hormone activated by the Hypothalamic-Pituitary-Adrenal (HPA) axis and is known to play a key role in the stress-related response and in the modulation of inflammation [49]. Under normal conditions, it serves a potent anti-inflammatory function and its circulating levels attenuate with negative feedback mechanisms. Yet, any dysfunctional response is likely to lead to unmodulated inflammation and it has been associated with pain hypersensitivity [50].

Some proponents claim that the mechanical stimulus provoked by SMT, typically associated with an audible cavitation, may trigger a cascade of neurophysiological responses orchestrated by the co-activation of the autonomous nervous system (ANS) and the HPA axis thus promoting tissue healing [51]. A critical evaluation of the evidence supporting immune-endocrine effects of SMT will follow in the next sections. 

### 4.2. Endocrine Markers

There are three studies [24,26,31] that have evaluated cortisol levels for immediate outcomes (up to 30 min) and two studies [26,31] for short-term (hours) follow-up time points. 

Plaza-Manzano et al. [26] conducted a study on 30 healthy subjects where they compared cortisol levels pre-, immediately post-SMT and 2 h post-SMT administered either to the cervical (*n* = 10) or to the thoracic (*n* = 10) spine in comparison with controls (*n* = 10) receiving venepuncture only.

They found that the interaction between SMT and cortisol levels accounted for 32% of the total variability. 

A significant increase of cortisol was found in the group receiving the cervical manipulation immediately post-intervention when compared with the thoracic group (mean difference, 4.10; 95% CI: 0.15–8.05; *p* < 0.040) and with the control group (mean difference, 4.60; 95% CI: 0.65–8.55; *p* = 0.018) respectively. The authors concluded that their results support the role of cervical, but not thoracic, HVLAT in cortisol secretion.

Yet, this study may be characterized by two possible flaws. Firstly, there are certain drawbacks associated with the use of venepuncture to extract blood samples as it is considered a procedure able itself to stimulate both the sympatho-adrenomedullary (SAM) system and the hypothalamus–pituitary–adrenal (HPA) axis [20,52]. Secondly, researchers investigated the effects of two different forms of manipulation. One group received a cervical manipulation involving rapid rotational forces of the head, whereas the second group was given a thoracic manipulation that did not involve head rotational movements. As such, the rise of cortisol immediately after the cervical manipulation, compared to the thoracic one, may be due to a disruption in the homeostasis of the vestibular system occurring in the former type of manipulation. Yet, more research is needed to support this hypothesis.

In this respect, an earlier study carried out by Christian et al. [24] reports evidence that SMT is not a stressful procedure. They observed no statistically significant differences in cortisol levels in a factorial 2 × 2 design, where subjects were allocated to four groups to receive either SMT or a sham manipulation and depending on whether they were symptomatic or not.

However, since each group received the intervention to multiple vertebral levels—including both cervical and thoracic ones—it is impossible to discern the unique contribution of cervical manipulation to HPA axis activation. In addition, the lack of randomization, allocation concealment, and participants blinding increased the risk of bias, thus casting doubts on the internal validity of this study. 

Whelan et al. [31] compared salivary cortisol profiles of three groups formed by 10 healthy individuals each, receiving a single cervical manipulation (CM, *n* = 10), a sham control involving manual contact and head positioning only (SHAM, *n* = 10), or no-intervention control (CTRL, *n* = 10). In contrast with the findings reported by Plaza-Manzano et al. [26], they found an attenuation of cortisol in all groups with no significant differences between the intervention group and controls.

Therefore, authors inferred that a cervical manipulation is not a stressful procedure because all groups followed a similar trend, and the overall decrease of cortisol was linked to physiological circadian rhythms. 

These rather contradictory results may be due to recruiting methods. Whereas Plaza-Manzano et al. [26] placed advertisements in a university and enrolled a mixed sample, Whelan et al. [31] recruited students of a chiropractic college who were familiar with the procedures. 

Another possible alternative explanation of these discordant findings—and an important aspect to consider when appraising studies on this topic—is the heterogeneity arising from the different type of cortisol collection procedures (salivary vs. venepuncture). Although salivary collection is considered a non-invasive approach for cortisol assessment which does not trigger the HPA axis, this method presents disadvantages related to participants’ adherence in self-collecting saliva samples [53]. Another aspect to consider, when interpreting the results of these studies, is the heterogeneity arising from the type of control procedures.

For instance, Plaza-Manzano et al. [26] compared two different types of manipulation as described above with an inactive control, Whelan et al. [31] used a sham procedure involving manual contact and spinal positioning without reaching a firm crisp end-feel and without high-velocity movements, and Christian et al. [24] differentiated the sham intervention from the “real” one by the application of a light pressure without evoking any audible cavitation.

There is an open debate as to what constitutes a placebo or sham intervention in the context of spinal manipulations [21,54]. As emphasized by Vernon et al. [54], a placebo HVLAT should ideally maintain features of indistinguishability and inertness, or at least of “structural equivalence” to keep participants blinded to the treatment they have been assigned to, thus minimizing non-specific, context-related effects.

Although these characteristics are crucial in clinical trial testing the effectiveness of spinal manipulation to rule out confounders, the extent to which a poorly designed placebo may affect the biological effects of SMT is unknown. 

A very recent study has demonstrated that non-specific factors, including verbal suggestions, can account for cortisol variations after SMT [35]. 

Investigators randomized 83 chronic neck pain patients (>3 months) in three groups receiving cervical manipulation (*n* = 28), a sham cervical manipulation (*n* = 28) and a cervical mobilization (*n* = 28). Prior to the delivery of the allocated technique, participants were given instructions with positive, neutral, or negative connotations (e.g., “This is a very effective intervention used to treat neck pain and we expect it to reduce your pain experience”). For the purpose of the analysis, they grouped participants depending on the instruction they randomly received (positive, neutral, and negative).

They found that irrespective of the manual therapy technique administered, those who were given positive instructions showed less salivary cortisol than the neutral (post-treatment mean difference: 0.11 (−0.26 to 0.05), *p* = 0.001) and negative (post-treatment mean difference: 0.25 (−0.41 to −0.09), *p* = 0.001) groups respectively. Interestingly, these differences in cortisol levels were not associated with a deterioration in pain intensity and disabilities. 

These results support the idea that even when investigating the biological effects of SMT, non-specific factors may play a determinant role in the modulation of the HPA axis.

Therefore, recruiting methods, stringent inclusion criteria, and well-designed sham manipulations are crucial to rule out contextual/psychological factors. Importantly, those who are naïve to manipulations might show augmented anticipatory activation of the stress pathways, especially when receiving a cervical manipulation related to other spine regions [21]. 

Earlier studies, not included in the abovementioned systematic review [19] because of low-quality, attempted to determine if SMT affects cortisol levels. Padayachy et al. [25] found that resting for 5 min in a supine position between the blood collection procedure and the spinal manipulation significantly lowered cortisol levels if compared with a group of similar symptomatic participants with no wait time following venepuncture. Considering that they recruited subjects with “mechanical”, acute lower back pain, it is not surprising that a longer period of rest in a recumbent position was perceived as less stressful, thus leading to diminished cortisol levels.

The study conducted by Tuchin [30] did not find significant differences in the proportion of cortisol after SMT. However, it is unclear whether participants were symptomatic or not, and it was not specified the nature and the area of application of the spinal manipulation. 

Subsequent to the publication of the systematic review by Kovanur-Sampath et al. [19], our electronic search retrieved further three studies [32,33,34]. A randomized controlled trial (RCT), conducted by Kovanur-Sampath et al. [32], found that a thoracic manipulation applied at the fifth thoracic vertebra of healthy men accounted for 28% of the variance of salivary cortisol levels [32]. Specifically, they found a statistically significant reduction of cortisol in the intervention group (*n* = 12) compared to the sham one (*n* = 12) at 5 min post-intervention (mean difference, 0.35; 95% confidence interval: (0.12, 0.6) *p* = 0.005), but not at 30 min and 6 h. These findings are in contrast with those found by Plaza-Manzano et al. [26] who observed an increase of cortisol immediately after the manipulation and a decline of cortisol short-term (2 h). Such inconsistency may be due to different methods in cortisol measurements and other unmeasured variables (e.g., temperature and humidity) that only Sampath et al. [32] attempted to control.

Valera-Calero et al. [34] analyzed salivary cortisol levels in a three-group RCT on patients suffering from neck pain lasting longer than 3 months. They compared the effects of a single cervical manipulation (*n* = 28, head rotation with a chin-hold technique) with cervical mobilization (*n* = 28, three sets of 1’ Grade III posterior to anterior with 1’ of rest in-between each set) and a sham manipulation (*n* = 27, head and hands positioning with no preload and thrust). The authors found that, although the within-groups analysis showed an increase of cortisol immediately both after the manipulation and the mobilization, the between-groups analysis returned no significant differences. Comparison of the findings with those of other studies confirms that, although a cervical manipulation (CM) seems able to trigger a stress response both in healthy [26] and pain patients [34] compared to a manipulation directed to the thoracic spine, CM does not appear to be more stressful than any other form of manual therapy technique directed to the neck. 

Lohman et al. [33] enrolled 28 female patients with non-specific mechanical neck pain (≤30 days symptoms duration) and divided them in two groups: one undergoing a bilateral cervical manipulation and a control group receiving a sham manipulation (hands positioning, but neither head movements nor thrusts). They evaluated cortisol levels through blood collection, which took place before each intervention and 20 s after their administration.

They found that the cortisol levels were not significantly different both within- and between- groups. In the discussion section, when comparing their results with previous research, the authors claim that their findings differ from those found by Plaza-Manzano et al. [26] because the latter investigators delayed blood samples collection. However, on close inspection, Plaza-Manzano and colleagues took blood samples immediately and 2 h after each intervention. It is more likely that such conflicting results are related to the different population examined (healthy vs. symptomatic).

Lohman et al.’s results mirror those of the previous studies that have examined the effects of spinal manipulation on symptomatic participants. All the three studies have found that cervical manipulations do not significantly affect cortisol levels on these types of populations. 

Collectively, all these findings provide mixed evidence for the endocrine effects of SMT. Overall, studies are of low quality, underpowered, and heterogeneous in terms of cortisol collection methods, types of intervention, and sham procedures adopted. Controlling covarying factors has been reported to be crucial to minimize confounding when measuring cortisol levels [55]. Indeed, it has been reported that a number of factors including, but not limited to, drinking/eating habits and physical activity levels, can alter the secretion of such hormone [56]. Additionally, the minimal clinical importance difference (MCID) for cortisol levels being currently unknown, the practical implications of these findings remain to be established. The lack of an accepted MCID could also have an impact on sample sizes calculation with an increased probability of type I or type II error occurrence [57]. 

Based on the arguments presented above, the existing evidence on the effects of spinal manipulation on cortisol levels should be taken with circumspection. Future high-quality studies with larger sample sizes and better-designed control groups may enhance our understanding and support plausible neuroendocrine mechanisms for the effectiveness of this therapeutic modality. 

### 4.3. Immune System Markers

Four studies investigated the proposition that SMT can exert effects on the immune system. Specifically, these studies investigated the effects of SMT on pro-inflammatory cytokines and humoral factors.

Teodorczyk-Injeyan et al. [28] measured the endotoxin-induced concentrations of TNF-α and IL-1β before and after the administration of three different procedures. One group (*n* = 24) received a thoracic spinal manipulation with an audible cavitation (SMT-C) delivered to a thoracic level detected as “restricted” by the clinician. The second group (*n* = 20) was given a similar hands-on procedure with altered parameters to avoid the acoustic component (SMT-NC), whereas the third group consisted of a venepuncture control (VC) group (*n* = 20) receiving manual contact only.

A significant decrease of in vitro TNF-α and IL-1β production was found in the SMT group only both at 20 min and 2 h after the manipulation, whereas both controls exhibited an increase of this cytokine at the same time interval.

Interestingly, these outcomes have not been paralleled by substance P (SP) variations, which is in contrast with the earlier findings reported by Brennan et al. [23] who found an elevation of both TNF-α and SP after a single thoracic manipulation.

It has been reported that a possible explanation for of such contradictory data between these two studies [23,28] may be due to different methods in culture systems (i.e., incubation period) and doses of the inducer (i.e., endotoxin) [58].

It is likely that the shorter incubation period (2 h) adopted by Brennan et al. [59], did not allow sufficient time for the neuroimmune reaction to subside, and hence the TNF-α content increased after SMT. 

Conversely, the longer period of incubation (24 h) used by Teodorczyk-Injeyan et al. [28], allowed the immunophysiological response to extinguish leading to a depletion of TNF-α.

The clinical relevance of these observations can be viewed from two different perspectives, depending on which effects of TNF-α prevail. It has been reported that this pleiotropic cytokine can play an ambivalent role in the modulation of inflammation, and opposite immunomodulatory effects under physiological and pathological conditions [60,61].

Firstly, SMT could trigger a self-limiting immune-related response that, in healthy individuals, may boost the beneficial effects of TNF-α leading to positive clinical outcomes.

Secondly, the activation of this cellular defense system may lead to the release of chemical constituents, which can act in the periphery by sensitizing nociceptors in the spinal cord by activating glial cells (microglia and astroglia) or in supraspinal centers by reaching the brain via the humoral and transport pathways [62].

It is likely that, in normal CNS sensitivity conditions, an increase of these signaling chemicals after SMT may lead to transient adverse effects (e.g., soreness, fatigue), whereas in circumstances of hyperexcitability, like in chronic pain states, such increase may drive to an escalation of events leading to pain exacerbation [58].

To further elucidate the mechanisms underpinning the effects of SMT on the immune system, Teodorczyk-Injeyan et al. performed two further analyses on the same set of participants [27,29]. They examined both the production of interleukin-2 (IL-2) and its modulation of the humoral immune response (antibody synthesis) by measuring the in vitro reactions of peripheral blood mononuclear cells (PBMCs) in culture supernatants elicited by different mitogens or inducers.

IL-2 has been reported to play a key role both in the initiation of the immune response by promoting the proliferation of antigen-stimulated T cells, and later in response to inhibit the immune response via pro-apoptotic effects [63,64]. In the earlier study [27], they found that IL-2 in vitro synthesis was significantly higher in those who received the thoracic manipulation with an audible cavitation compared to venepuncture control at both 20 min (*F* = 14.30, *p* = 0.000) and 2 h after the manipulation (*F* = 12.99, *p* = 0.001). Interestingly, the sham manipulation group followed a similar trend with significant differences in IL-2 concentrations compared to venepuncture control at both 20 min (*F* = 8.01, *p* = 0.006) and 2 h after the manipulation (*F* = 9.54, *p* = 0.003).

In the subsequent analysis [29], they found that SMT can increase the in vitro production of mediators of the humoral immunity (immunoglobulin G and immunoglobulin M), in response to exogenous IL-2 induction. However, such positive findings may be somewhat biased by the fact that blood preparations of 11 subjects were not analyzed due to insufficient numbers of PBMCs.

Moreover, the study design adopted by Teodorczyk-Injeyan et al. [27,28,29] does not allow speculation as to whether the same positive results would have been achieved if manipulations were applied to “unrestricted” spinal levels or to alternate segments. Although an audible cavitation of the joint is considered a prerequisite for a “successful” HVLAT manipulation in clinical trials [14], it is unclear whether the cavitation is necessary for the activation of the immune response. This is partially confirmed by the fact that both the group receiving the index HVLAT and the group receiving the sham one followed similar trends. 

Future studies using a factorial design may help to gain a better understanding of the specific effects of SMT on the immune system.

In summary, the current literature identifies a potential role of SMT in modulating the immune response. However, the clinical relevance remains mostly unanswered because most of the studies have been conducted on healthy subjects. This is significant because it is unclear how the elements of the innate and adaptive immunity would function after SMT under pathological circumstances. To further complicate the picture, recent studies revealed contradictory data regarding the relationship between levels of pro-inflammatory chemicals and pain manifestations [65,66]

## 5. Conclusions

Spinal musculoskeletal (MSK) disorders have an outsized impact on healthcare systems. Despite the significant advancements in our understanding of these conditions, effective interventions for the treatment of lower back and neck pain are lacking and they represent a burden especially in modern industrialized societies. Although accumulating research suggests that psychosocial factors are strong predictors of pain chronicity, the role that the biological domain may play on patients’ symptoms should not be ignored altogether. Tissue modifications, low-grade inflammation, and a dysfunctional stress-related system response seem to mutually affect each other and are well-established features of these MSK conditions. Spinal manipulation therapy (SMT) is an intervention widely administered by many healthcare professionals to provide pain relief and improve functionality. The shift in perspective regarding the analgesic mechanisms underlying SMT, moving from an entirely mechanistic paradigm to a holistic one, has led researchers to investigate the effects of spinal manipulations on different neuro-immune-endocrine biomarkers. 

Although it has been demonstrated that SMT provides short-term benefits across different spinal MSK disorders, the available evidence supporting the capacity of SMT to trigger a significant immune-endocrine response is mixed and its clinical relevance remains to be established. Quality issues, small sample size, lack of studies on symptomatic subjects, and heterogeneity related to methods of biomarkers collection and sham procedures limit the interpretation of findings. Further high quality and adequately powered studies are needed to draw valid inferences on the biological plausibility of SMT and to support its consistent implementation in clinical practice.

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
