# Peer review of "The Effects Induced by Spinal Manipulative Therapy on the Immune and Endocrine Systems"

_medicina, 2019, doi:10.3390/medicina55080448_

Round 1

Reviewer 1 Report

I congratulate authors for their complete and detailed review on effects of spinal manipulative therapy on endocrine and immune systems. They have done extensive search on studies related to their review topic and found few relevant articles. They have discussed these studies outcomes and the shortcomings of some of them on their assumptions and investigations have been addressed in this review. This helps readers to determine where more investigation about this topic is needed. So, I believe, this review is useful and valuable. However, authors do not clearly address all these relevant papers in their review. For example, in “Abstract” part and in line 23, they mention that their electronic search retrieved thirteen relevant articles while in line 110 of the “Result” part, they mention the number as fourteen. Line 24 to 27 of “Abstract” is exactly like line 111 to 114 of the “Results”. I do not see any advantages in repeating few lines twice in two parts of the paper. It is good idea authors highlight those relevant articles by referring to them. For example: Ten studies have investigated the effect of SMT on cortisol levels. (Line 112). Authors can put references in brackets and highlight which articles those are. I know authors have mentioned some of those articles in “Discussion” part, but I believe it is still not clear enough for readers.

By the way, it is good idea that all relevant articles are discussed in this review as the number of them is small (only thirteen or fourteen). For example, the manual search yielded three additional manuscripts. (line 110). They are references 22-24. I do not see anywhere in this paper, reference 23 has been discussed.

In line 19, word HVLAT has been used while it has been defined for the first time at line 61. I would suggest authors define it first time they use it.

Line 71 can be shifted to References part

The “Results” part is only the result of the authors electronic search not the result of review itself. I believe if authors combine results and discussion in one title, the paper will look more professional.

Thank you.     

Author Response

Response to Reviewer 1 Comments

We are glad you have found our work of interest and we would like to thank you for your kind feedback. We very much appreciate your time in reviewing our manuscript and your suggestions to improve its quality.

Point 1: in “Abstract” part and in line 23, they mention that their electronic search retrieved thirteen relevant articles while in line 110 of the “Result” part, they mention the number as fourteen.

Response 1: We have checked and amended the number of papers we retrieved through our electronic and hand search and we apologise for the confusion. 

Point 2: Line 24 to 27 of “Abstract” is exactly like line 111 to 114 of the “Results”. I do not see any advantages in repeating few lines twice in two parts of the paper. It is good idea authors highlight those relevant articles by referring to them. For example: Ten studies have investigated the effect of SMT on cortisol levels. (Line 112). Authors can put references in brackets and highlight which articles those are. I know authors have mentioned some of those articles in “Discussion” part, but I believe it is still not clear enough for readers.

Response 2: We have added further details and references to the result section in order to help readers understanding which articles we are referring to.

Point 3: By the way, it is good idea that all relevant articles are discussed in this review as the number of them is small (only thirteen or fourteen). For example, the manual search yielded three additional manuscripts. (line 110). They are references 22-24. I do not see anywhere in this paper, reference 23 has been discussed.

Response 3: All the relevant references and included studies are now within the discussion section. We believe the previous incongruity was related to references enumeration. 

Point 4: In line 19, word HVLAT has been used while it has been defined for the first time at line 61. I would suggest authors define it first time they use it.

Response 4: the acronym HVLAT has been defined and written in its non-abbreviated form.

Point 5: Line 71 can be shifted to References part

Response 5: Line 71 has been referenced properly.

.

Point 6: - The “Results” part is only the result of the authors electronic search not the result of review itself. I believe if authors combine results and discussion in one title, the paper will look more professional.

Response 6: The "Results" section has been moved under the "Discussion" one and the section enumeration has been amended accordingly.

We hope we have satisfied the reviewer’s concerns and amended the manuscript properly. Should you have any further request please do not hesitate to contact us.

Thank you

Kind regards

Reviewer 2 Report

I really enjoyed reading this work. Excellent work describing very precisely the mechanisms that SMT impacts the cortisol release with exact time point and thus the immune system, which are very possible. Every part of this review is well written without any need for more explanation.

Please make only one correction in the abstract: explain the  abbreviation "HVLAT".

Author Response

Dear Reviewer 2,

Thank you very much for your kind feedback and we feel flattered that you enjoyed our work.

thank you

kind regards